# Arthroscopic Debridement Enhanced by Intra-Articular Antibiotic-Loaded Calcium Sulphate Beads for Septic Arthritis of a Native Knee Following Iatrogenic Joint Injection: A Case Report

**DOI:** 10.3390/medicina60101636

**Published:** 2024-10-07

**Authors:** Simone Alongi, Elisa Troiano, Cristina Latino, Giovanni Battista Colasanti, Tommaso Greco, Carlo Perisano, Massimiliano Mosca, Stefano Giannotti, Nicola Mondanelli

**Affiliations:** 1Department of Medicine, Surgery and Neurosciences, University of Siena, 53100 Siena, Italy; 2Orthopedic Unit, Azienda Ospedaliero-Universitaria Senese, 53100 Siena, Italy; 3Orthopedics and Trauma Surgery Unit, Department of Ageing, Neurosciences, Head-Neck and Orthopedics Sciences, Università Cattolica del Sacro Cuore, Fondazione Policlinico Universitario A. Gemelli IRCCS, 00136 Rome, Italy; 4Second Clinic of Orthopaedics and Traumatology, IRCCS Istituto Ortopedico Rizzoli, 40136 Bologna, Italy

**Keywords:** septic arthritis, arthroscopy, calcium sulphate beads, reabsorbable pearls, antibiotic carrier, intra-articular injection, native joint

## Abstract

Septic arthritis (SA) represents an orthopedics urgency and mainly affects the knee joint. Due to its devastating effects on cartilage, immediate management is crucial. SA is characterized by an annual incidence of 2 to 10 cases per 100,000 individuals, with mortality rates fluctuating between 0.5% and 15%, with a substantially higher mortality rate observed in older people (15%) in contrast to younger cohorts (4%). The etiology of septic arthritis is multifactorial: a spectrum of Gram-positive and Gram-negative bacteria can contribute to the development of this condition, especially *Staphylococcus aureus*. The treatment involves urgent (arthroscopic or arthrotomic) debridement associated with adequate antibiotic therapy. Intra-articular antibiotic carriers can also be used to increase their local concentration and effectiveness. The case of a 67-year-old woman affected by knee SA from methicillin-susceptible *S. aureus* is presented. She was treated with an arthroscopic debridement enhanced by intra-articular antibiotic-loaded calcium sulphate beads, together with antibiotic therapy. At 2-year follow up, the infection had been eradicated and the patient fully recovered. This is the first description, to our knowledge, in the English literature, of the use of antibiotic-loaded calcium sulphate beads as an adjuvant in the surgical treatment of SA of a native knee joint.

## 1. Introduction

Acute septic arthritis (SA) is a rare but serious orthopedic surgical urgency that can lead to severe cartilage destruction and serious life-threatening complications [1]. It occurs when bacteria invade the synovium and joint space, followed by an inflammatory process. Pro-inflammatory molecules, such as cytokines, interleukins, and proteases, released into the articular chamber, mediate damage to the cartilage up to its destruction. Generally, SA patients present with acute monoarticular joint pain, swelling, fever, warmth, limited range of motion, limp, and pseudoparalysis. The annual incidence of this pathology has been estimated to be 2 to 10 cases per 100,000 individuals [2], with mortality ranging from 0.5% to 15%, depending on the involved bacterium, promptness of treatment, and patient age (15% in the elderly versus 4% in the young) [3]. An exception are children, especially under the age of two, who have a higher incidence of SA in large joints [4]. 

The most frequently affected joint is the knee which is involved in about 50% of the cases [1,3], followed by the hip, shoulder, elbow, and ankle, although any synovial joint could become infected [3,5]. Most of these infections reach the joint through hematogenous dissemination; contiguity from adjacent infectious foci or iatrogenic inoculation are also possible, other than open wounds and penetrating traumas. Iatrogenic SA has been reported in patients after hyaluronic acid (HA) and/or glucocorticoids (GCs) intra-articular injection, and its risk has been estimated to be around 0.0002% to 0.072% [5]. *Staphylococcus aureus* is the most frequently encountered pathogen, but other Gram-positive and Gram-negative bacteria may be involved in the development of SA [6]. The prevalence and susceptibility of organisms that cause SA have not altered substantially during the past decades [7]. Pre-existing joint disease is the strongest risk factor for SA, but systemic chronic diseases, such as diabetes, immunodeficiency states, or skin conditions, as well as traumatic or recent surgical intervention, can also play a role in the development of SA [6,8].

Early detection and correct diagnosis of SA is fundamental for significantly improving prognosis and quality of life. A delay in treatment could lead to major complications, like cartilage compromission, osteonecrosis, chronic osteomyelitis, chronic pain, leg length discrepancies, sepsis, and death [1,2,3,8,9]. Diagnosis of SA is based on a patient’s history and examination, and it is confirmed with imaging, laboratory exams, and by identification of the infecting agent in the synovial fluid. Some differential diagnoses should be excluded, such as crystal-induced arthritis, inflammatory arthritis, osteoarthritis, neoplastic diseases, or adverse reaction to injection therapy. The finding of a white blood cell count higher than 1100/microL with at least a 64% neutrophil population in the synovial fluid represents a strong indicator of SA [10]. Instrumental tests complete the investigation: traditional radiographs are generally unable to provide information in the early stages; if there is advanced structural damage, a widening of the joint space or structural changes in the subchondral bone may become evident. Ultrasound is used as a first approach, and it can be useful to guide an arthrocentesis [11]. At the same time, magnetic resonance imaging (MRI) can detect bone and soft tissue alterations in the early stage and identify and quantify the degree of cartilaginous involvement [12]. The test that still represents the gold standard in diagnosing SA is the microbial culture of a synovial fluid sample. In some cases, the pathogen has intrinsic characteristics that make it difficult, if not impossible, to culture through insemination or has a growth rate incompatible with timely treatment. Nowadays, it is possible to use more sophisticated methods, such as protein chain reaction or metagenomics with next-generation sequencing techniques to identify the agent [13,14]. Anyway, antimicrobial susceptibility testing is not feasible with such techniques, to date, and an antibiogram on cultures remains the gold standard.

The principle of treatment of SA is intravenous antibiotic therapy combined with immediate surgical treatment: without culture tests, it is advisable to administrate empirical, broad-spectrum antibiotic therapy. Anti-staphylococcal coverage should always be considered for any age and risk category [9]; Vancomycin, Nafcillin, and Oxacillin are the primary choices. The treatment must be modified accordingly as soon as an antibiogram is available. Surgical debridement and irrigation to reduce the intra-articular bacterial load should be performed immediately [1,2]. Also, surgical debridement will help antibiotics to arrive at the infected joint with bleeding. Despite the lack of consensus on the ideal surgical approach – arthrotomic or arthroscopic – for this kind of surgery, recent studies have shown similar efficacy in infection eradication, but an earlier functional recovery and a lower complication rate when considering the arthroscopic approach [1,3,15]. An important tool in choosing the most suitable surgical treatment is represented by the Gächter classification (Table 1) [2,16,17]; it is based on the evaluation of the visual characteristics of the joint aspirate and the presence/absence of structural alterations detectable through imaging. According to Gächter classification, only stage IV should be treated with open surgery; this occurs in less than 10% of cases. Furthermore, it plays a fundamental prognostic role in predicting a patient’s outcome, as it also involves an evaluation of the arthroscopic findings [18]. The first phase of the arthroscopic procedure is the evacuation of infected fluids and an intense lavage with almost 10 liters of saline solution. Then, debridement of infected or severely damaged soft tissue is mandatory [18]. A solution of 1% methylene blue dye could be injected intra-articularly before beginning the debridement procedure. In fact, it has been demonstrated that methylene blue preferentially stains not only necrotic host tissue but infected and inflammatory tissue as well [19]. Therefore, during debridement, it helps the surgeon to spare healthy tissue that should be preserved, as the synovial membrane represents an immunocompetent structure and a natural barrier. Arthroscopic therapy is highly effective in treating knee joint infections, with an overall healing rate of 90–100% [18]. Concurrently, the intra-articular use of antibiotic-loaded calcium sulphate beads has been extensively studied in periprosthetic joint infections and osteomyelitis to ensure a locally high and long-lasting antibiotic concentration [20,21]. However, to date, no report of the use of antibiotic-loaded calcium sulphate beads for bacterial SA in a native joint, treated with open surgery or arthroscopically has been found in the literature. Based on the quantity of calcium sulfate used and the molecules added, the time necessary for complete reabsorption of the carrier varies from 3 to 12 weeks [20].

In the present report, the case of a 67-year-old woman affected by iatrogenic knee SA from methicillin-susceptible *S. aureus* (MSSA) is presented, who was treated with arthroscopic debridement enhanced by intra-articular antibiotic-loaded calcium sulphate beads and associated with adequate antibiotic therapy, with a 2-year follow up. At our institution, no Ethical Committee nor Institutional Review Board approval is necessary for case report studies, and the patient gave her informed consent to data collection and their anonymous use for scientific and teaching purposes. This study was performed in accordance with the ethical standards as laid down in the 1964 Declaration of Helsinki and its later amendments or comparable ethical standards.

## 2. Case Presentation

A 67-year-old woman presented to our attention in the emergency department with a painful and swollen left knee, lasting for 2 days. She previously underwent GCs plus HA injection into her left knee for symptomatic treatment of mild primary knee osteoarthritis. Her past medical history was silent except for pain and functional limitation in her left knee. The procedure was performed elsewhere one week before she arrived at our institution. In the days following the procedure, the patient developed signs of inflammation (*tumor, rubor, calor*) with increasing pain (*dolor*) and functional limitation (*functio laesa*). A color doppler ultrasound of the lower limb was performed, yielding negative results for phlebitis or deep venous thrombosis. Subsequently, a knee arthrocentesis was performed, yielding turbid and orange-colored synovial fluid, which was sent for microbial culture. Blood test showed high inflammatory indices (erythrocyte sedimentation rate, C-reactive protein, fibrinogen, D-dimer), while urate levels were in the normal range. Empiric oral antibiotic therapy with amoxicillin/clavulanic acid and sulfamethoxazole/trimethoprim was initiated, pending consultation with an infectious disease specialist, and surgical debridement was scheduled. The Knee injury and Osteoarthritis Outcome Score (KOOS) was administered to the patient, asking her to fill it twice, referring to the last week’s symptoms and the actual ones. A significant worsening in the KOOS was observed in relation to the usual symptoms: the score dropped from 86% to 30%. Using the Gächter classification as a guide, the surgical procedure was performed arthroscopically, with the patient in a supine position, under spinal anesthesia, following the administration of two grams of pre-operative intravenous cefazolin. After carefully washing the limb with povidone–iodine antiseptic solution, the operating field was set up. During the surgical procedure, before opening arthroscopic portals, synovial fluid was collected for a new microbiological culture (Figure 1). Through the same access, methylene blue dye was injected (Figure 2); thereafter, anteromedial, anterolateral, and superolateral arthroscopic portals were created.

After prolonged lavage to clear the joint from turbid synovial fluid and the dye that both impeded complete vision, initial intra-articular inspection was performed. Significant synovitis and diffuse uptake of the previously instilled dye were observed, especially in soft tissues, such as the suprapatellar pouch, anterior compartment, intercondylar notch, and gutters. The arthroscopic findings, in association with the quality of the synovial fluid collected, were consistent with stage III of the Gäcther classification. Soft tissue was also collected for microbiology culture (three samples), and accurate synovectomy and removal of all colored soft tissues was performed. Particular attention was given to the preservation of the cartilage, menisci, and ligaments structures (Figure 3).

At the end of the procedure, 5 mL of reabsorbable calcium sulphate beads (Stimulan^®^ Rapid Cure, Biocomposites Ltd., Keele, UK) supplemented with 500 mg vancomycin and 160 mg gentamicin, mixed according to the manufacturer’s specific instructions, were instilled through the arthroscopic cannula from the superolateral portal (Figure 4). Antibiotics were chosen for their broad-spectrum coverage, while waiting for the microbial culture with the antibiogram. No drainage was placed into the joint in order to maintain the concentration of antibiotics released by the calcium sulfate pearls as high as possible. The choice to use different antibiotics in systemic and local treatment was taken both because not all molecules can be added to the calcium sulfate and to obtain a synergistic effect, offering the patient the best possible antibiotic coverage while waiting for the antibiogram.

Post-operative antibiotic therapy was continued with the same regimen until isolation of a MSSA from the initial microbial culture on day 4 after collection (Figure 5). Therefore, based on the antibiogram, a switch to intravenous levofloxacin and fosfomycin was initiated on the recommendation of the infectious disease specialist. Cultures from intra-operative material returned negative results; a new consultation with an infectious disease specialist was performed, and fosfomycin was replaced with daptomycin. The patient was then discharged, and antibiotic therapy was prolonged for six weeks. The administration of intravenous daptomycin took place at patient’s home with the territorial continuity of nurse care service, while levofloxacin was self-administered orally. The subsequent outpatient ambulatorial follow up visits were carried out in collaboration with infectious disease specialists. At 2-weeks follow up, a drastic reduction in the inflammation indices was already observed; the patient did not report any feverish episodes, and the physical examination showed a significant improvement with the initial recovery of joint function (KOOS, 70%). 

In subsequent follow ups, physical examination showed constant improvement with the progressive recovery of range of motion and decreasing blood inflammatory indices until normalization (Figure 6); the disappearance of painful symptoms was also reported. 

The patient resumed weight-bearing walking on the operated limb without restrictions. At 1-month follow up, the patient was in excellent health; she had also almost completely recovered her joint function, and only a sensation of muscular tension remained (KOOS 80%). Knee radiographs were taken, and they showed the complete reabsorption of the antibiotic carrier, in the absence of significative structural alterations that could be considered direct or indirect signs of infection (Figure 7). At 2-years follow up, the patient was in excellent health; she did not report any inflammatory symptoms, and she did not complain of functional deficits or pain other than anterior knee pain (KOOS 85%), consistent with her osteoarthritis (Figure 8).

## 3. Discussion

Acute SA of a native joint is one of the most critical orthopedic urgencies. It can affect all synovial joints, although data show that about 50% of cases involve the knee, and *S. aureus* is the most common pathogen. The physiopathology of joint damage is multimodal, with both infection itself and joint inflammatory response representing the key elements of it. Particular attention must be paid to all procedures that require access to the articular chamber, mainly if performed outside a sterile setting, since a communication is created between the synovial cavity and the external environment. Among these procedures, viscosupplementation with HA and GCs injections are a common practice in the conservative management of symptomatic osteoarthritis. However, intra-articular injections, as well as all diagnostic and therapeutic practices that involve piercing the skin, can bring microorganisms from the skin into the joint, contaminating it and eventually leading to SA. SA of a native joint represents the most dangerous complication of intra-articular injection, despite a reported low incidence rate (as “higher” as less than 1/1,000) [5]. SA can cause enormous joint damage, which can lead to irreversible alterations of the joint structures with consequent disability, even leading to the patient’s death by sepsis. Early identification and appropriate treatment—consisting of combined surgical debridement and systemic antibiotic therapy—are of paramount importance [1,2,5]. Nevertheless, to date, there is no clear consensus on the optimal therapeutic approach, and several surgical options had been successfully applied [1,2]. A timely and accurate synovial debridement aiming to remove as much infected material as possible is key to success, and it can be performed with both arthrotomic or arthroscopic techniques [1,2,3,15]. Recent studies have shown similar effectiveness of open surgery and arthroscopically procedure in eradicating an infection of a native knee, even though arthroscopic treatment was associated with lower complication rate and better functional recovery [1,2,3,15]. To obtain the most accurate debridement possible, a coloring method was employed to showcase the infected tissues. Methylene blue has demonstrated its ability to stain infected, nonviable, and inflammatory host tissues, and it has been used as an effective debridement guide in the intra-operative setting [19,23].

One of the alternate diagnoses to be aware of, in the context of an acute painful and swollen knee, is a gout flare; often, initial symptoms are similar between SA and gout. An accurate anamnestic collection regarding the presence of similar episodes in the past, family history, and patients’ comorbidities, is fundamental. Imaging is extremely nonspecific: the typical periarticular “punch” erosions with protruding edges are present in only 45% of gout cases and after 6 to 12 years from the first acute episode. However, there are specific diagnostic criteria for gout [24,25,26], that help in confirming it as a diagnosis. On the other hand, blood urate levels in normal range or a history of recent intra-articular maneuvers support the diagnosis of acute SA. In both cases (suspected gouty access and SA), it is advisable to perform an arthrocentesis with microbiological culture and chemical–physical examination. Another possible differential diagnosis is the immune reaction to the substance, which can also occur after multiple intra-articular infiltrations [27,28,29]; acute manifestations after administration of intra-articular substances are not rare, with a frequency of 1–2 cases per 100,000 infiltrations [27]. The literature has coined the following definition of pseudosepsis [30]: “severe inflammation of the joint often with significant cellular effusion and significant pain, normally occurring within 24 and 72 hours after intra-articular injection”. This condition is not self-limiting and it requires surgical treatment similar to that of SA. A higher incidence rate of this complication is linked to the formulation of the used HA, especially when cross-linked formulations are preferred in which the combination of the high permanence of the compound in the joint and the presence of degradation products of the drug combine to the persistence of a potentially immunogenic inflammatory stimulus; this nevertheless happens with avian-derived formulas [28,29]. However, pseudosepsis remains a diagnosis of exclusion today after septic arthritis and gout. Recent studies conducted by Dragomir et al. [29] have highlighted the importance of individual susceptibility and the remarkable similarities with immuno-serological profiles of rheumatoid arthritis in which the following are preponderant: an increase in the pro-inflammatory interleukins 1β and 6, myeloperoxidase, and activation of the complement system; cases where fragments of HA are found within some macrophage populations [29]. Tracing the GCs and HA type in the reported case was not possible since the patient was not been primarily treated at our hospital.

The concept of topical antibiotic therapy appeared before the 1970s when molecules of different antibiotic concentrations were locally administered “naked”. This concept was taken up and developed with the studies of Buchholz and Engelbretch [31] on the use of antibiotic-loaded polymethylmethacrylate (PMMA) in the treatment of periprosthetic joint infections (PJIs), and then further evolved with calcium-based carriers, hydroxyapatite, or bioactive glass [32,33]. The local use of calcium sulfate spheres loaded with antibiotics has been widely studied for the prevention and treatment of multiple infectious conditions, including osteomyelitis and PJIs, but also for soft tissue diseases or preventive purposes in risky situations [20,34,35]. However, the indications for its use have expanded to also include post-operative infections following closed and open fractures, spontaneous bone infections, and native joint infections [21]. The choice of calcium sulfate carrier, in our practice, is dictated by therapeutic needs and its versatile characteristics. These include a known elution profile, its reabsorbability, but after an adequate residence time, ability to fill the dead space, and the possibility of being used in beads of different dimensions or as bullets or a paste [33]. The device’s manageability has proved a winning feature, allowing its intra-articular administration via arthroscopic means. Although antifungal-impregnated PMMA beads have been described as a viable therapy for fungal septic arthritis of a native joint [36], to date and to our knowledge, the present is the first case of arthroscopically allocated antibiotic carriers and the first case of reabsorbable antibiotic carrier used in a SA of a native joint reported in the literature. Unlike previous-generation substances, calcium-based carriers can be added to various molecules, even in combination; they guarantee higher concentrations of local antibiotics, avoiding drops in concentrations below the therapeutic threshold for the entire period of their stay. With a 100% release of antibiotic and without exposing the organism to systemic administration, calcium sulfate carriers significantly reduce the risks associated with classic antibiotic therapy, such as hepato- or nephro-toxicity, metabolic deactivation from the hepatic passage before it is delivered where it is needed, and reduced concentration on site, providing a safer alternative [20,33,37]. A further reason for choosing these substances in cases of SA is the ability to actively act against biofilm, a structure which, in its mature phase, thanks to a multi-layered structure, presents an antibiotic resistance 10,000 times higher than the free-floating planktonic microorganisms. Calcium-based carriers allow adequate minimum inhibitory concentration and minimum biofilm eradication concentration values to be achieved and maintained for several weeks [32,33,38]. An important aspect to consider is the risk of “third body wear” which has been widely studied with respect to prosthetic joints. In these cases, the use of reabsorbable calcium sulfate beads in a prosthetic joint seems to be safe with respect to subsequent wear. On the other hand, sensible doubts are present with respect to cartilage damage in a native joint. Anyway, when calcium sulfate beads are inserted into a joint, they occupy the free space they found (in a knee: the intercondylar notch, the patellar pouch) and, therefore, the risk of scratching the cartilage is reduced. Also, the beads are reabsorbable and in contact with body fluid they may soften and molder if pressed. Moreover, cartilage damage due to persistent infection can balance the risk of iatrogenic damage due to third body wear of the beads. The literature available regarding the use of antibiotic carriers in cases of periprosthetic infection is much more extensive, both in cases of debridement and implant retention and debridement antibiotic pearls and retention of the implant, as well as in one-stage or two-stage revisions. Certainly, the criteria for choosing the most suitable carrier are different, having to evaluate mechanical resistance, formation of microfractures in the material, and articulation and mobility. Of particular importance in this case are the formation of bacterial biofilm on the prosthetic components and third-body wear. However, their use remains debated, as can be seen from other studies [39].

In the present case report, in our opinion, the best available techniques were employed. First, the most extensive debridement was obtained with the help of methylene blue guide. Secondly, the arthroscopic technique was employed to reduce complication rate and improve functional results. Lastly, a higher and long-lasting local antibiotic concentration was achieved with the use of reabsorbable antibiotic carrier.

Limitations are present in the present paper. This is a case report, and generalizations are obviously not possible. Also, plain radiographs were the only imaging performed at follow up, but the patient was feeling well and she refused to take a control MRI. On the other hand, the present case is, to our best knowledge, the first report about the use of a reabsorbable antibiotic carrier in a native joint, and the first report about the arthroscopic placement of such a device.

## 4. Conclusions

SA of a native joint represents a clinical condition that may lead to long-term complications up to joint destruction, and it must be treated as soon as possible. A correct diagnosis is of paramount importance, yet it could be difficult. Other conditions may mimic it, such as acute episodes of gout or an adverse reaction to HA injection. SA requires urgent surgical treatment by the orthopedic surgeon, but also a multidisciplinary approach is mandatory to obtain the best result. The administration of calcium sulfate beads charged with antibiotics can help in obtaining the eradication of the infection, and it is feasible during an arthroscopic procedure. Moreover, it seems not to mechanically compromise articular cartilage. In our experience, this surgical technique, associated with accurate and thorough debridement and systemic antibiotic therapy, has proven effective in eradicating an MSSA infection of a native knee.

## Figures and Tables

**Figure 1 medicina-60-01636-f001:**
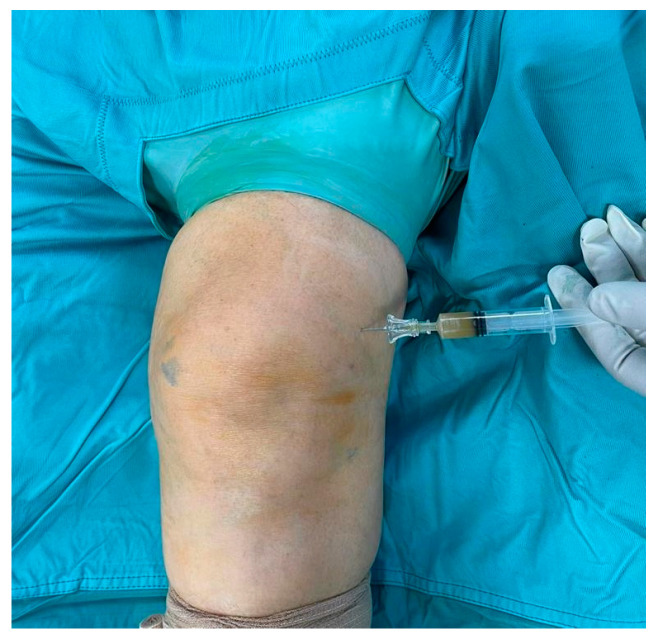
Intra-operative arthrocentesis.

**Figure 2 medicina-60-01636-f002:**
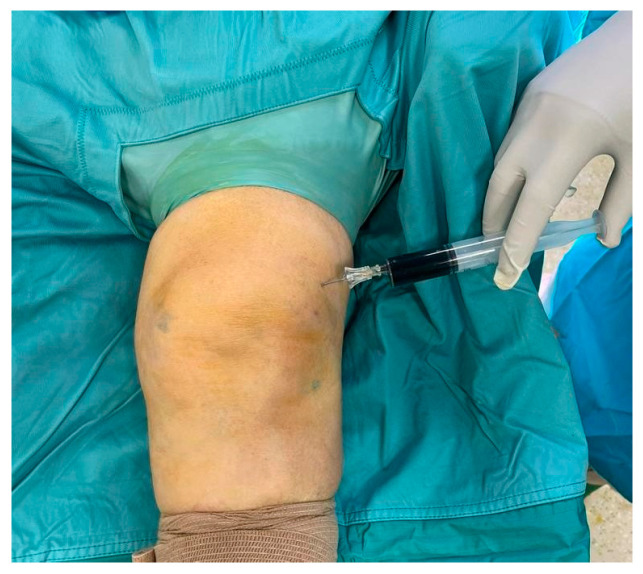
Intra-articular injection of 10 mL of diluted 0.1% methylene blue solution.

**Figure 3 medicina-60-01636-f003:**
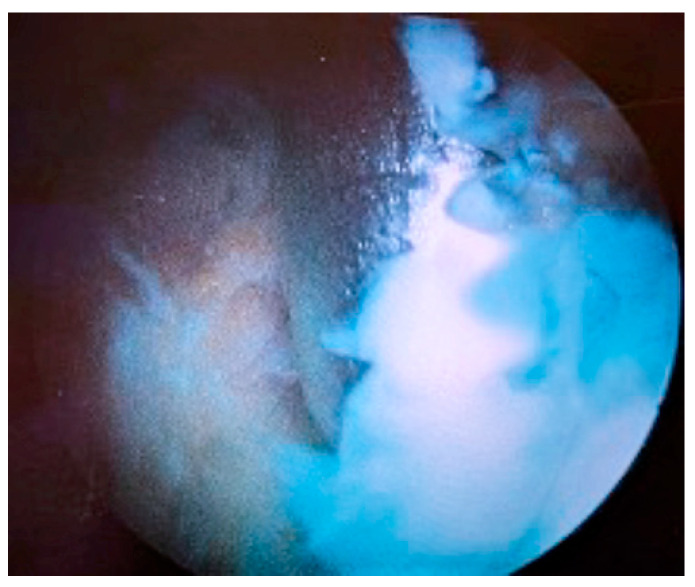
Arthroscopic synovectomy.

**Figure 4 medicina-60-01636-f004:**
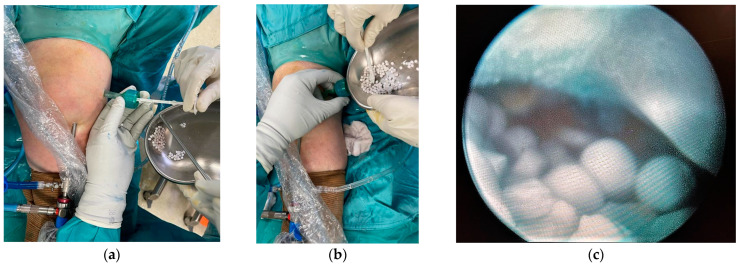
(**a**) Intra-articular introduction of Stimulan^®^ beads through the arthroscopic cannula at the superolateral portal, (**b**) using a proper pusher. (**c**) Intra-articular vision of the beads.

**Figure 5 medicina-60-01636-f005:**
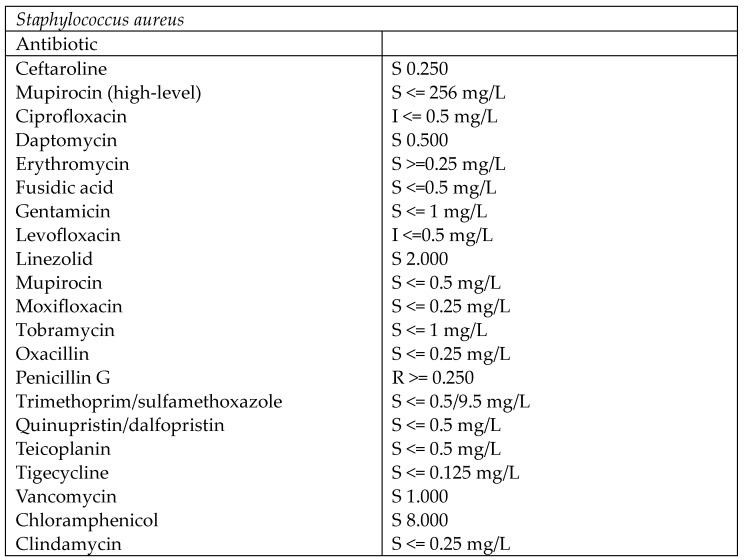
Antibiogram of joint fluid sample.

**Figure 6 medicina-60-01636-f006:**
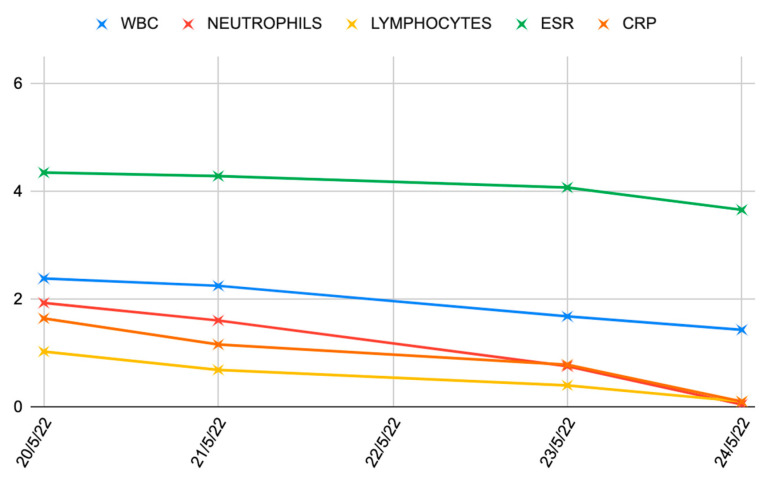
Blood inflammatory index trend (represented in logarithmic scale). ESR: erythrocyte sedimentation rate, CRP: C-reactive protein.

**Figure 7 medicina-60-01636-f007:**
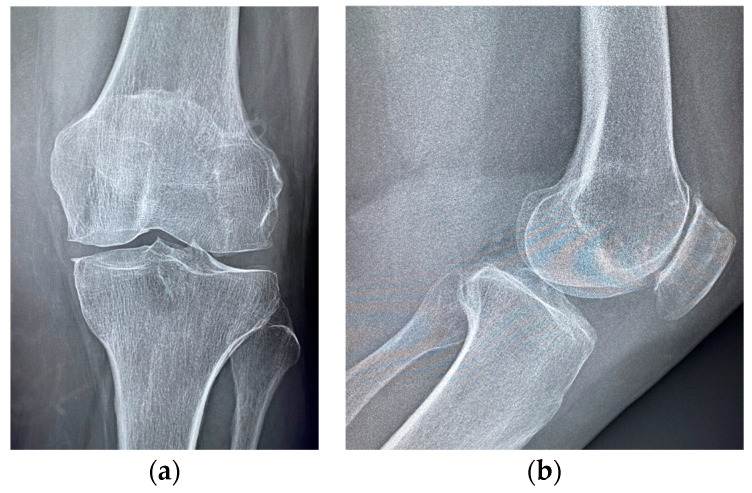
Knee radiographs at 1-month follow up; (**a**) anteroposterior and (**b**) lateral view.

**Figure 8 medicina-60-01636-f008:**
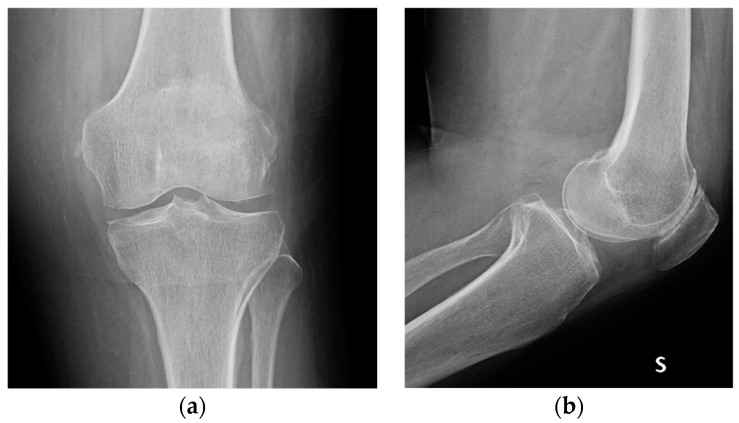
Knee radiographs at 2-years follow up, showing only minimal patello-femoral osteoarthritic progression with respect to immediate post-operative radiographs; (**a**) anteroposterior and (**b**) lateral view.

**Table 1 medicina-60-01636-t001:** The Gächter classification of SA of a native joint, re-edited from Stutz et al. [16,22].

Stage I	Opacity of fluid, redness of the synovial membrane, possible petechial bleeding, no radiological alterations
Stage II	Severe inflammation, fibrinous deposition, pus, no radiological alterations
Stage III	Thickening of the synovial membrane, compartment formation (“sponge-like” arthroscopic view, especially in the suprapatellar pouch), no radiological alterations
Stage IV	Aggressive pannus with infiltration of the cartilage, possibly undermining the cartilage, radiological signs of subchondral osteolysis, possible osseous erosions and cysts

## Data Availability

The datasets used and/or analyzed during the current study are available from the corresponding author on reasonable request.

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
