# Peer review of "Arthroscopic Debridement Enhanced by Intra-Articular Antibiotic-Loaded Calcium Sulphate Beads for Septic Arthritis of a Native Knee Following Iatrogenic Joint Injection: A Case Report"

_medicina, 2024, doi:10.3390/medicina60101636_

Round 1

Reviewer 1 Report (Previous Reviewer 3)

Comments and Suggestions for Authors

I thank the authors of the manuscript “Arthroscopic debridement enhanced by intra-articular antibiotic-loaded calcium sulphate beads for septic arthritis of a native knee following iatrogenic joint injection: a case report.” for considering my comments and remarks in the previous review when I got the opportunity to look into the paper. The authors have removed some comments and provided an antibiogram to prove the effectiveness of re-absorbable calcium sulfate beads in the described case, and added a graphical illustration of the trend of inflammatory markers and an assessment of the patient's condition on an international scale “Knee injury and Osteoarthritis Outcome Score” (KOOS) before and after treatment, which is highly appreciated.

Still, in terms of the language, the manuscript calls for further reviewing concerning grammar, sentences’ structure, punctuation, and Italian words instead of English; also, the layout needs editing in places, e.g. lines 24, 35, 50, 114, 117, 134, 141, 195, 201-205, 222, 249. Please, convert Figure 5 into a table.

Comments on the Quality of English Language

The manuscript calls for further minor reviewing concerning grammar, sentences’ structure, punctuation, and Italian words instead of English.

Author Response

Reviewer 2 Report (New Reviewer)

Comments and Suggestions for Authors

The authors have reported a case of “Arthroscopic debridement enhanced by intra-articular antibiotic-loaded calcium sulphate beads for septic arthritis of a native knee following iatrogenic joint injection”.

The following comments and suggestions are provided with the aim of improving the quality of the manuscript and increase its interest for the readers.

1. Introduction

Lines 68-69. Ultrasound is used as a first approach, and it can be useful to guide an arthrocentesis.

Comment: no reference is provided regarding the findings on US indicating or suggesting infection

Lines 69-70. At the same time, magnetic resonance imaging (MRI) can detect bone and soft tissue alterations in the early stage and identify and quantify the degree of cartilaginous involvement [11,12].

Comment: The two references cited do not provide specific criteria for MRI suggesting infection. Moreover, the involvement of the articular cartilage would not qualify for early diagnosis, limited to the involvement of the synovial membrane. A recent reference which might be appropriate is the following: “Usefulness of MRI findings in differentiating between septic arthritis and transient synovitis of hip joint in children: A systematic review and meta-analysis” (2022).

Lines 71-72. The test that still represents the gold standard in diagnosing SA is the microbial culture of a synovial sample.

Comment: It is not always the case. Early diagnosis might rely on culture of the synovial fluid only, as in the reported case with synovial culture after the start of the antibiotic therapy, if needed.

Lines 84-85. If no improvement is seen after 5-6 days, performing a new arthrocentesis is

advisable to exclude fungal infections or reactive arthritis.

Comment: Non-infectious forms of arthritis as well as fungal infections should be excluded before starting antibiotic therapy in the differential diagnosis.

Lines 91-92. An important tool in choosing the most suitable surgical treatment is represented by the Gächter classification (Table 1) [16,17].

Comment: The Gächter classification was published in 1993 (ref 17), when MRI technology was not comparable with the present one and most probably needs to be updated and revised.

Lines 101-104. During debridement, it helps to discriminate between necrotic tissues

and tissues exposed to the infection, that should be debrided, and more deep tissues that

did not come into contact with the microbial agent and viable tissues, that should be preserved as the synovial membrane represents an immunocompetent structure and a natural

barrier.

Comment: It is unclear the definition of “necrotic tissue” in septic arthritis, which is usually edematous with increased capillary proliferation and acute inflammatory infiltrate with loss of surface layer and fibrinous exudate, but no transmural necrosis. It is also unclear which “viable tissue or tissue not in contact with the microbial agent(s) should be preserved. It is suggested to revise the period accordingly to the comment.

Based on the quantity of calcium sulfate used and the molecules added, the time necessary for complete reabsorption of the carrier varies from 30 to 60 days [22].

Comment: The resorption of the beads of calcium sulfate of the reference cited [22] is in bone tissue and not synovium. Therefore, the resorption rate might not be comparable as well as the effect.

2. Case presentation

Line 131. Her past medical history was silent.

Comment: The patient must have had prior symptoms for the steroid and hyaluronic acid injection.

Lines 136-137. A knee arthrocentesis was performed yielding turbid and orange-colored synovial fluid which was sent for microbial culture.

Comment: The authors have the choice to include differential diagnosis in this section or in the discussion. Proper consideration should have been given to adverse reaction to injection to hyaluronic acid with appropriate references and laboratory findings. Moreover, results of the microbiology culture should be reported and if negative, the rationale of starting empiric antibiotic therapy.

Lines 143-144. A significant reduction was observed in relation to the usual symptoms: the score dropped from 86% to 30%.

Comment: The authors should comment why, with such a dramatic result in only one week of treatment, they decided to do synovial debridement instead of waiting for possible resolution of the symptoms monitoring through repeated laboratory testing.

Lines 160-161. After prolonged lavage to clear the joint from purulent synovial fluid and the dye

that both impeded a correct vision

Comment: The fluid that was turbid has become purulent after starting empirical antibiotic therapy seems improbable and the description needs to be consistent in both reports.

Lines 165-167. Soft tissue was also collected for microbiology culture, and accurate synovectomy and removal of all colored soft tissues was performed.

Comment: An explanation why the tissue was not submitted for histological examination should be provided. Additionally, how many samples were sent for microbiology culture (one or more).

Lines 188-189. Post-operative antibiotic therapy was continued with the same regimen until isolation of the MSSA from the initial microbial culture.

Comment: It looks that MSSA is mentioned for the first time; if so, complete spelling of the acronym is advised. Additionally, the day of culture positivity after collection should also be provided.

3. Discussion

Lines 260-261. Methylene-blue has demonstrated ability to stain bacterial biofilm and nonviable host tissues.

Comment: It is unclear how the bacterial biofilm, if present in the synovium, might be stainable with methylene blue. The biofilm indicated in the references provided refers to periprosthetic joint infection (PJI) which is not applicable to the reported case.

Lines 297-300. Compared to other substances, such as PMMA, for which microscopic analyses have highlighted a statistically significant relationship with the appearance of signs of wear on the contact surfaces, the use of calcium sulfate-based carriers does not appear to be subject to this phenomenon [35–37], with respect to a metallic surface.

Comment: The concept of third body wear refers to joint implants only. In this case, it should refer to possible interposition of the beads between two articular surfaces with damage of the cartilage surface layer or deeper, in case of presence of a variable degree of osteoarthritis, indicated by the need of steroid and hyaluronic acid injection. The authors should clarify the concept only for the reported case.

Lines 317-318. In the present case report, in our opinion, the best available techniques were exploited.

Comment: Suggested “employed” instead of “exploited”. The comment on the techniques should be limited to the use of intraarticular antibiotic beads as a novelty.

4. Conclusion

Comment: The conclusion should also include also the difficulty of the differential diagnosis with adverse reaction to hyaluronic acid, even if for this case it was not properly considered.

Comments on the Quality of English Language

Minor English language editing has been provided in the comments.

Author Response

This manuscript is a resubmission of an earlier submission. The following is a list of the peer review reports and author responses from that submission.

Round 1

Reviewer 1 Report

Comments and Suggestions for Authors

This case report is based on a single case study reporting the effectiveness of antibiotic-loaded calcium sulfate beads as an adjunctive therapy for septic arthritis (SA) in native joints.

Comment 1. Please provide a detailed explanation in the discussion section why antibiotic-loaded beads were chosen for the treatment of native joint SA, particularly in a typical case scenario. Clearly outline the anticipated benefits of using antibiotic-loaded beads, emphasizing how they are expected to improve treatment outcomes compared to conventional therapies.

Comment 2: It would be valuable to include a discussion on existing studies regarding the use of antibiotic-loaded beads in periprosthetic joint infections (PJI) due to the controversy and varing outcomes reported in the literature.

Author Response

This case report is based on a single case study reporting the effectiveness of antibiotic-loaded calcium sulfate beads as an adjunctive therapy for septic arthritis (SA) in native joints.

Comment 1. Please provide a detailed explanation in the discussion section why antibiotic-loaded beads were chosen for the treatment of native joint SA, particularly in a typical case scenario. Clearly outline the anticipated benefits of using antibiotic-loaded beads, emphasizing how they are expected to improve treatment outcomes compared to conventional therapies.

Reply: Added.

Comment 2: It would be valuable to include a discussion on existing studies regarding the use of antibiotic-loaded beads in periprosthetic joint infections (PJI) due to the controversy and varing outcomes reported in the literature.

Reply: A hint to the use of antibiotic-loaded beads in PJI has been added. However, we believe that deepening the topic would not be of interest regarding this specific case report since the local environment of a native joint significantly differ from that of a prosthetic one.

Reviewer 2 Report

Comments and Suggestions for Authors

Good manuscript, needs correction.

1. Ethical approval- even though it is just a case report, it is better to get ethical approval from other parties like the Ministry of Health.

2. The consent form should be included in the appendix.

3. Figure 3 is not clear, better change it.

4. Line 283 - error in the wording

5. Discussion on the other materials used by other literature should be included to check and compare your current method.

6. Include the limitation of the study.

7. Ref no 16, line 359 - please double check.

8. Ref no 11 - please double check.

Comments on the Quality of English Language

Minor grammatical error

Author Response

Good manuscript, needs correction.

Ethical approval- even though it is just a case report, it is better to get ethical approval from other parties like the Ministry of Health.

Reply: As we mentioned in the manuscript, at our Institution no Ethical Committee nor Institutional Review Board approval is necessary for case report studies. Therefore, no approval request can be submitted as it would be deemed not necessary for such manuscript nor for the treatment.

The consent form should be included in the appendix.

Reply: We included the informed consent for treatment and for publication into the editorial manager procedure; as the name of the patient and her signature could be recognizable, they should not be included into public materials; anyway, the editor have a copy of them.

Figure 3 is not clear, better change it.

Reply: Done.

Line 283 - error in the wording.

Reply: Corrected

Discussion on the other materials used by other literature should be included to check and compare your current method.

Reply: Done.

Include the limitation of the study.

Reply: Limitations have been added.

Ref no 16, line 359 - please double check.

Reply: Corrected with the English paper from the same Author.

Ref no 11 - please double check.

Reply: Corrected.

Reviewer 3 Report

Comments and Suggestions for Authors

Acute septic arthritis, though being a rare orthopedic surgical urgency, is one of the most critical and injures the knee in approximately 50% of cases. The work “Arthroscopic debridement enhanced by intra-articular antibiotic-loaded calcium sulphate beads for septic arthritis of a native knee following iatrogenic joint injection: a case report.” describes a case of a patient with knee septic arthritis successfully treated with an arthroscopic debridement enhanced by intra-articular antibiotic-loaded calcium sulphate beads, together with antibiotic therapy. It is worth noting that according to the information provided in the article, as a result of the therapy, the patient ultimately did not have any complaints of diseases of the knee joint after 2 years and fully recovered.

Nevertheless, there are some points in the manuscript to be clarified.

1. It is necessary to provide an antibiogram to prove the effectiveness of re-absorbable calcium sulfate beads in the described case.

2. In my opinion, it is necessary to present the dynamics of changes in the leukogram in order to be able to assess the systemic inflammatory response.

3. It would be useful to present an assessment of the patient's condition on an international scale before and after treatment.

4. It would be interesting to compare the described case with similar cases in your hospital when other treatment regimens were applied.

5. I would like to draw the authors attention to some issues concerning the English language. Firstly, to some state expressions, for example, lines 117-118; 176; 188; 227. Sometimes, passive voice should be used instead of active (e.g. line 128). Listings should be as follows: one, second, and third. A comma should be put before and (lines 133, 138, 165, and many more). Also, the whole manuscript should be revised for correct use of articles, plurals/singulars, and prepositions.

Comments on the Quality of English Language

I would like to draw the authors attention to some issues concerning the English language. Firstly, to some state expressions, for example, lines 117-118; 176; 188; 227. Sometimes, passive voice should be used instead of active (e.g. line 128). Listings should be as follows: one, second, and third. A comma should be put before and (lines 133, 138, 165, and many more). Also, the whole manuscript should be revised for correct use of articles, plurals/singulars, and prepositions.

Author Response

Acute septic arthritis, though being a rare orthopedic surgical urgency, is one of the most critical and injures the knee in approximately 50% of cases. The work “Arthroscopic debridement enhanced by intra-articular antibiotic-loaded calcium sulphate beads for septic arthritis of a native knee following iatrogenic joint injection: a case report.” describes a case of a patient with knee septic arthritis successfully treated with an arthroscopic debridement enhanced by intra-articular antibiotic-loaded calcium sulphate beads, together with antibiotic therapy. It is worth noting that according to the information provided in the article, as a result of the therapy, the patient ultimately did not have any complaints of diseases of the knee joint after 2 years and fully recovered. Nevertheless, there are some points in the manuscript to be clarified.

It is necessary to provide an antibiogram to prove the effectiveness of re-absorbable calcium sulfate beads in the described case.

Reply: The antibiogram has been added to the manuscript as figure 5.

In my opinion, it is necessary to present the dynamics of changes in the leukogram in order to be able to assess the systemic inflammatory response.

Reply: A graphical illustration of the trend of inflammatory markers has been added to the manuscript as figure 6.

It would be useful to present an assessment of the patient's condition on an international scale before and after treatment.

Reply: The patient was assessed with KOOS (Knee injury and Osteoarthritis Outcome Score) before admission and at each follow-up visit. Information has been added in the manuscript.

It would be interesting to compare the described case with similar cases in your hospital when other treatment regimens were applied.

Reply: This is a very interesting topic, and we are planning to deepen it in the next future. However, this is the first case of a septic arthritis on a native joint in adulthood treated at our hospital, so we are not able to compare our results to other treatment regimens. This may be explained by two reasons. Septic arthritis is a rare entity, and as a referral hospital of the region, we usually treat highly complex pathologies such as PJIs.

I would like to draw the authors attention to some issues concerning the English language. Firstly, to some state expressions, for example, lines 117-118; 176; 188; 227. Sometimes, passive voice should be used instead of active (e.g. line 128). Listings should be as follows: one, second, and third. A comma should be put before and(lines 133, 138, 165, and many more). Also, the whole manuscript should be revised for correct use of articles, plurals/singulars, and prepositions.

Reply: The manuscript has been thoroughly revised and English language has been corrected.

Comments on the Quality of English Language: I would like to draw the authors attention to some issues concerning the English language. Firstly, to some state expressions, for example, lines 117-118; 176; 188; 227. Sometimes, passive voice should be used instead of active (e.g. line 128). Listings should be as follows: one, second, and third. A comma should be put before and (lines 133, 138, 165, and many more). Also, the whole manuscript should be revised for correct use of articles, plurals/singulars, and prepositions.

Reply: Done.